# Relationship Between Early Childhood Non-Parental Childcare and Diet, Physical Activity, Sedentary Behaviour, and Sleep: A Systematic Review of Longitudinal Studies

**DOI:** 10.3390/ijerph16234652

**Published:** 2019-11-22

**Authors:** Silvia Costa, Sara E. Benjamin-Neelon, Eleanor Winpenny, Veronica Phillips, Jean Adams

**Affiliations:** 1UKCRC Centre for Diet and Activity Research, MRC Epidemiology Unit, School of Clinical Medicine, University of Cambridge, Cambridge CB2 0QQ, UK; sara.neelon@jhu.edu (S.E.B.-N.); ew470@medschl.cam.ac.uk (E.W.); jma79@medschl.cam.ac.uk (J.A.); 2School of Sport, Exercise and Health Sciences, Loughborough University, Loughborough LE11 3TU, UK; 3Department of Health, Behavior and Society, Johns Hopkins Bloomberg School of Public Health, Johns Hopkins University, Baltimore, MD 21205, USA; 4Medical Library, School of Clinical Medicine, University of Cambridge, Cambridge CB2 0SP, UK; vmp26@cam.ac.uk

**Keywords:** early childhood, early care and education, dietary behaviours, activity behaviours, obesity risk factors

## Abstract

The rising prevalence of childhood obesity is a global public health concern. Evidence suggests that exposure to non-parental childcare before age six years is associated with development of obesity, diet, and activity behaviours (physical activity, sedentary behaviour, and sleep). However, findings are inconsistent and mostly from cross-sectional studies, making it difficult to identify the direction of causation in associations. This review identified and synthesised the published research on longitudinal associations between non-parental childcare during early childhood, diet, and activity behaviours. Seven databases were searched, and results were independently double-screened through title/abstract and full-text stages. Included studies were evaluated for risk of bias. Of the 18,793 references screened, 13 met eligibility criteria and were included in the review. These presented results on 89 tested childcare/outcome associations, 63 testing diet outcomes (59% null, remainder mixed), and 26 testing activity behaviour outcomes (85% null, remainder mixed). The scarce available literature indicates little and mixed evidence of a longitudinal association. This reflects a paucity of research, rather than clear evidence of no effect. There is an urgent need for studies investigating the longitudinal associations of non-parental childcare on diet and activity behaviours to assess potential lasting effects and mechanisms; whether and how effects vary by provider; and differences by intensity, duration, and population sub-groups.

## 1. Introduction

Globally, in 2018, approximately 40 million (6%) children under 5 years of age were overweight or obese [1]. Obesity during childhood is associated with increased risk of both obesity and a range of other conditions later in life, including low self-esteem, high blood pressure, insulin resistance, coronary heart disease, and stroke [2,3,4]. The early years (<6 years of age) have been repeatedly highlighted as a critical period for the development and prevention of obesity [3,5,6], as well as the establishment of related healthy habits, such as healthy diet, physical activity, and sleep patterns [7,8]. Several individual, inter-personal, and environmental factors influence the development of childhood obesity [9]. Because they affect large numbers of children, environmental factors such as childcare settings represent potential targets for obesity prevention [10,11]. This is reflected in guidance and policies in some countries on physical activity and food in some childcare settings [12,13,14].

An increasing number of children now attend non-parental childcare prior to age 6 years, and many spend large proportions of their week days in such care [15,16]. A report by the United Nations’ Children’s Emergency Fund shows that around 80% of 3–6 year olds and 25% of 0–3 year olds in developed countries spend time in some form of childcare [16]. A growing body of research suggests that attendance at childcare is associated with increased adiposity or risk of obesity in children [17,18,19]. However, the available evidence is inconsistent [17,18,20,21], and may partly depend on different aspects of the childcare received, such as the type (i.e., informal or formal care) or intensity (e.g., number of hours per week).

The ways in which non-parental childcare might affect obesity are poorly understood [21,22]. Different types and characteristics of childcare settings may have different influences on the development of obesity-related risk factors [11,23,24,25,26]. Evidence suggests that some types of non-parental childcare (e.g., grandparents or Head Start in the United States (US)) and staff behaviours (e.g., giving non-food rewards and allowing children to self-serve) are associated with diet patterns and behaviours [25,27]. Similarly, different types (e.g., home-based versus centre-based settings) and features (e.g., staff behaviours like playing with children) of childcare are associated with physical activity [28,29,30,31] and sedentary behaviour [24,31,32] in young children. There is also some evidence that attending some types of childcare is associated with problematic sleeping patterns in young children [23,26,33]. However, the direction of these associations is mixed, and associations are not consistently found in all population sub-groups or studies. Additionally, the vast majority of the current evidence comes from cross-sectional studies, making it difficult to determine the direction of causation.

The aim of this review was to systematically gather and synthesise the published research on the longitudinal relationship between non-parental childcare in the early years and diet, physical activity, sedentary behaviour, and sleep. We focused exclusively on longitudinal studies to increase confidence that any association we find might be causal.

## 2. Methods

This review was part of a larger programme of reviews (including obesity and stress outcomes alongside the diet and activity behaviour outcomes reported here) that was registered with the PROSPERO database (registration number CRD42015027233) [34], and is reported in line with the Preferred Reporting Items for Systematic Reviews and Meta-analysis (PRISMA) recommendations [35]. The protocol for the overall programme of systematic reviews has been published elsewhere [36].

### 2.1. Search Strategy

Seven electronic bibliographic databases were searched in January 2016, using a predefined search strategy: MEDLINE, EMBASE, PsycINFO, Web of Science, Scopus, Applied Social Sciences Index and Abstracts (ASSIA), and the Scientific Electronic Library Online (SciELO). Searches were restricted to human subjects, but there were no restrictions placed on publication date or language. The search strategy was informed by search strategies of relevant previous systematic reviews [37]. An experienced university librarian (V.P.) reviewed the search strategy, adapted it for different databases, and ran the searches. An example of the search strategy used for the MEDLINE and Embase databases can be found in Appendix A. Results were managed using EndNote^®^ software. The searches were re-run at the end of May 2017.

### 2.2. Study Selection and Eligibility Criteria

After removal of duplicates, records were screened in two phases using a pre-piloted procedure (Figure 1). In phase one, title and abstracts were screened by two reviewers working independently against the phase one eligibility criteria described in Figure 1. The full texts of all studies identified by either reviewer as potentially eligible were retrieved. In phase two, full texts were screened by two reviewers working independently against the phase two eligibility criteria described in Figure 1. In cases of uncertainty or discrepancy between reviewers, we consulted a third reviewer and consensus was achieved by discussion.

The number of papers included and excluded at each stage of the review process is shown in Figure 2. The diagram shows overall number of references screened and excluded (by reason) instead of number of references by outcome of interest, because eligibility at full-text screening (phase two) was done for all six outcomes of interest in the registered protocol for the programme of systematic reviews (obesity, stress, diet, physical activity, sedentary behaviour, and sleep) [34]. Concurrently, the final number of included studies are presented for behaviour outcomes only that were the focus of the current review—diet, physical activity, sedentary behaviour, and sleep.

Details of and justification for each eligibility criteria are described in full in the protocol [36]. Studies were included where participants were children aged <6 years and not in primary school at first assessment, and living in middle- and high-income countries as defined by the World Bank [38]. Only observational longitudinal study designs, including case-control, prospective, and retrospective designs, were included. The exposure of interest was non-parental childcare where there was between-child variation in exposure, for example by timing of attendance (i.e., age when care started and stopped), intensity (i.e., full- or part-time care), duration (i.e., years of childcare), types (i.e., formal or informal; private or public), or simply attendance versus non-attendance. Studies were included where outcomes were objectively assessed or proxy/self-reported measures of diet, physical activity, sedentary behaviour, or sleep. Studies were excluded if they were not published in peer-reviewed journals.

Study authors were contacted via email for clarification of any issues not clear to the reviewers. If authors did not reply by the end of the data extraction stage, studies were excluded. Conference abstracts, masters and doctoral theses were excluded, as these do not necessarily go through a formal peer-review process. Nevertheless, the authors of any potentially relevant records of these types were contacted via email to determine if peer-reviewed journal articles had resulted, and these were screened as above.

### 2.3. Data Extraction and Management

A standardised and pre-piloted form was used to extract data from included studies for assessment of study quality and evidence synthesis. This captured information about study setting and population, exposure and outcome variables, statistical analyses and results, as stated in the protocol [36]. The first author extracted these data into an Excel^®^ database, and a second author (J.A./S.B.N.) independently checked the extracted information against the full-texts of included studies.

### 2.4. Data Synthesis

Key information was tabulated (e.g., sample characteristics, exposure, and outcome measures) for each study, grouped by outcome variable, and a narrative synthesis of the included studies performed. Because of heterogeneity in exposure and outcome variables, meta-analysis was not appropriate. This also meant that it was not possible, as originally planned, to perform a quantitative synthesis of differences in effect between different types and features of childcare, different outcomes, high- and middle-income countries, ages at exposure, and socio-demographic sub-groups (e.g., by ethnicity). Instead, and because of the sometimes large number of relevant exposure and outcome variables used in included studies, all individual relevant associations reported in the included studies were included here.

### 2.5. Quality Assessment

An adaptation of the United States Department of Agriculture’s Nutrition Evidence Library Bias Assessment Tool (NEL-BAT) [39] was used to assess risk of bias in included studies. This tool assesses risk of selection, performance, detection, and attrition bias. For observational studies, possible scores range from 0 to 26, with lower scores indicating lower risk of bias. S.C. and J.A. independently assessed all included studies for risk of bias, and disagreements in scores were resolved by discussion. 

## 3. Results

We identified 47,529 articles. After de-duplication, 18,793 articles underwent title/abstract screening, and the full texts of 175 articles were reviewed. Thirteen studies [40,41,42,43,44,45,46,47,48,49,50,51,52] met all of the eligibility criteria, and were included in the review. Of these, eight studies reported on diet [40,46,47,48,49,50,51,52], three reported on physical activity [40,42,45], three on sedentary behaviour [42,43,44], and one on sleep [41] outcomes. Some studies reported on more than one of diet, physical activity, sedentary behaviour, and sleep.

### 3.1. Summary of Included Studies

A detailed description of each study’s characteristics can be seen in Table 1. Most included studies were from high-income countries, with seven originating from the United States [40,41,43,44], one from Australia [42], one from New Zealand [45], and one from the United Kingdom (UK) [49]. Samples were generally balanced with relation to children’s gender (although three studies did not report gender composition of the sample) [48,49,52], but varied greatly both in size (between 34 and 18,050 subjects) and ethnic composition (between 0% and 87% white, with one study not reporting race/ethnicity or country of birth [42], and five providing information only for country of birth) [46,47,49,50,52]. 

All studies assessed exposure to non-parental childcare between birth and 5 years via proxy-report from a parent or primary caregiver. Nine studies described childcare exposure in simple categorical terms (e.g., centre-based preschool or Head Start centre versus other, including parents) [40,41,42,46,47,49,50,51,52]. One study assessed duration of exposure (centre-based preschool for at least 2 years versus other/mixed care, including parents) [40]. Five studies assessed intensity of non-parental childcare (e.g., average number of hours in childcare per week) [43,44,45,48,49].

Age at outcome assessment varied from 1–12 months to 51 years. For diet outcomes, all studies used proxy-report by a mother or main caregiver [40,46,47,48,49,50,51,52]. For physical activity outcomes, most studies used proxy- or self-report [40,42,45], with average accelerometer counts/minute used in one case [45]. All studies assessing sedentary behaviour outcomes using proxy-report by a parent. For sleep, quantitative outcomes (e.g., nap durations) were measured objectively by an accelerometer, whereas qualitative outcomes (e.g., difficulty going to bed or falling asleep) were subjectively measured by parent-report [41]. 

No two studies used the same outcome variables. All diet studies presented outcomes as categorical variables, with three studies investigating breastfeeding-related outcomes [48,49,52], four studies investigating consumption of specific foods identified as healthy or unhealthy [40,46,47,51], and one study investigating between-meal eating [50]. Two studies presented physical activity outcomes as categorical variables (e.g., high, medium, and low physical activity level versus sedentary) [40,42], whereas one study used continuous variables (e.g., average accelerometer counts per minute). All sedentary behaviour variables were categorical (e.g., >4 versus ≤4 h/day of television (TV) viewing), and two out of three studies used TV viewing as a proxy for sedentary behaviour. There was a wide range of sleep variables, from number and duration of naps to variables relating to the quality of sleep.

Four studies investigated only one exposure and one outcome variable [42,43,44,48]. The remainder explored several outcome [41] or exposure variables [40,45]. Thus, the included studies reported on 63 associations between non-parental childcare and diet outcomes, nine associations with physical activity outcomes, three associations with sedentary behaviour outcomes, and 15 associations with sleep outcomes. 

### 3.2. Synthesis of Findings

Table 2 presents detailed results for all relevant associations explored in each study.

### 3.3. Diet

Eight studies evaluated the longitudinal relationship between non-parental childcare and diet outcomes [40,46,47,48,49,50,51,52]. Results were highly mixed. Lee et al. [47] reported that children who attended Head Start settings at 4 years of age showed significantly higher frequency of healthy eating patterns at 5–6 years of age than those attending other settings (all *p* < 0.05), except pre-kindergarten. Conversely, no differences in frequency of unhealthy eating patterns were found between the groups [47]. Another study assessing attendance at Head Start [40] found that children who attended Head Start or other centre-based childcare at 4 years of age (irrespective of length of exposure) were more likely to report frequent fruit consumption than those in other/mixed care (including parental care) at age 5–6 years. In this study, children who attended other (non-Head Start) centre-based childcare were also less likely to regularly consume soda at age 5–6 years than those in other/mixed care (all *p* < 0.05). However, other centre-based childcare was also associated with higher likelihood of regular consumption of chips (*p* < 0.05). There was no difference in the likelihood of regular consumption of fast-food, candy, and chips consumption, and frequent consumption of vegetables between those attending Head Start or other centre-based childcare (irrespective of length of exposure) versus other/mixed care (including parental care). 

Similarly, Wasser et al. [51] also reported mixed findings. They found that children in any non-maternal childcare had higher odds of consuming whole fruit (odds ratio (OR): 1.15, *p* < 0.05), and juice (any childcare OR: 1.51; grandparents OR: 1.91, *p* < 0.05) than those in maternal care. But there was no association between childcare (overall or by type) and consumption of five other food and drinks, including vegetables and salty snacks. Camara et al. [46] also reported mostly null associations between childcare and the two dietary patterns investigated, except a higher adherence to a processed/fast-food pattern at 2–5 years of age in those being cared for at home by someone other than the mother compared to those cared for by their mother (B = 0.70 (SE: 0.14), *p* < 0.001) at 2–3 years of age. Sata et al. [50] reported more frequent between-meal eating before dinner at age 6 years in those cared for by grandparents and nursery/kindergarten than those cared for by mothers, as well as between meal eating ≥3 times per day those cared for by grandparents versus those cared for by mothers at 3 years of age. However, no other associations were found with any care at 12 or 22 years of age. 

Three studies investigated breastfeeding outcomes [48,49,52], showing mixed results. Pearce et al. [49] reported lower likelihood of breastfeeding for ≥4 months in children attending informal compared to parental care (independent of attending full- or part-time, lone parenthood, or area of deprivation), but mixed results for those attending formal care. For example, in the analyses stratified by family structure, children living in single parent families receiving formal care were more likely to be breastfed for ≥4 months (risk ratio (RR) = 1.65) than those being cared for by parents, but the reverse was true for children living in couple families (RR = 0.79, all *p* < 0.05). While Weile et al. [52] reported a higher risk of changing from mostly breastfed to mostly or solely formula-fed in those attending childcare compared to those cared for at home (RR = 2.05 to 2.50, *p* < 0.05), Levy et al. [48] found an increased risk of earlier cessation of breastfeeding in children who used a pacifier and did not attend any childcare and those who attended 15 days of childcare between 0–6 months of age versus those not attending childcare and not using a pacifier.

Overall, the eight included studies tested 63 associations between non-parental childcare exposures and diet outcomes. Of these, 37 (59%) were null, 10 (16%) indicated significant beneficial effects of non-parental care on dietary behaviours, seven (11%) indicated significant detrimental effects of non-parental care on dietary behaviours, one (2%) reported a significant association with between-meal snacking (but it is not clear if this was conceived as positive or negative for health) [50], and eight (13%) found mixed results. As an example of the latter, Pearce et al. [49] reported that the effect of informal care on length of breastfeeding varied by parental educational attainment. 

### 3.4. Physical Activity

Three studies evaluated the longitudinal relationship between childcare during early childhood and physical activity outcomes [40,42,45]. Results were highly mixed. Belfield and Kelly [40] found that children who attended Head Start at 4 years old had significantly lower physical activity levels in kindergarten than those who received parental care. However, there was no difference in physical activity between those attending other centre-based care versus parental care, irrespective of length of exposure to such care. Conversely, D’Onise et al. [42] reported that those attending Kindergarten Union preschool between 2 and 4 years old were more likely to be in the high physical activity level group (versus sedentary group) at around age 45 years than those who did not attend this preschool. In the only study looking at intensity of childcare use, Taylor et al. [45] found no significant associations between weekly hours of childcare attendance at 3 or 4 years old and objectively measured physical activity 1 or 2 years later. 

Overall, the six included studies tested nine associations between non-parental childcare exposures and physical activity outcomes. Of these, seven (78%) were null [40,45], whereas two (22%) found significant differences but in competing directions [40,42].

### 3.5. Sedentary Behaviour 

Three studies evaluated the longitudinal relationship between childcare during early childhood and sedentary behaviour outcomes [42,43,44], including one study that conceptualised sedentary behaviour as the absence of physical activity (also reported on above) [42]. As noted, D’Onise et al. [42] reported that those who attended Kindergarten Union preschool between ages 2 and 4 years were less likely to be in the sedentary group (versus the high physical activity level group) at around 45 years, than those who did not attend this preschool. The remaining two studies [43,44] found no significant associations between number of hours per week of childcare at 24–36 months and 3–5 years and subsequent daily hour of television viewing at 36 months and 6–12 years, respectively. 

Thus, three associations between non-parental childcare and sedentary behaviour were tested in included studies. Two (67%) were null [43,44], and one (33%) showed a significant association between childcare and lower risk of sedentary behaviour [42].

### 3.6. Sleep

The only study investigating the longitudinal relationship between childcare and sleep outcomes yielded mixed results [41]. Cairns and Harsh [41] reported that those attending all day preschool or daycare at age 5 years (versus a primary/secondary caregiver) transitioned to earlier sleep onset and wake up time on week days in the first months of preschool. The authors are clear that the health implications of these differences are unknown. There were no differences between the groups in any other variables (e.g., difficulty in going to bed and nocturnal sleep duration on week days). 

Overall, 15 associations between non-parental childcare and sleep outcomes were tested. The majority (*n* = 13) were null, with only two showing significant results. The health implications of these were not clear.

### 3.7. Quality Evaluation

Risk of bias scores ranged from 1–12 out of 26 (with lower scores indicating lower risk of bias) (see Table 3). The most common sources of bias were not reporting or using valid and reliable outcome measures (12 studies), and outcome assessors not blinded (or not clear whether they were blinded) to the intervention or exposure status of participants (10 studies). There was low risk of bias throughout in terms of inclusion and exclusion criteria, recruitment strategy, accounting for variations in the execution of the study from the proposed protocol or research plan, and length of follow-up across study groups.

## 4. Discussion

### 4.1. Summary of Findings

To our knowledge, this is the first systematic review to investigate the longitudinal relationship between non-parental childcare before age 6 years and diet, physical activity, sedentary behaviour, and sleep. Overall, the evidence base is very limited, with only 13 studies meeting our eligibility criteria. Eight studies reported on diet outcomes, three on physical activity, three on sedentary behaviour, and one on sleep. Included studies varied widely in terms of definition and measurement of both exposure and outcomes, and lacked in-depth exploration of different aspects of childcare that may influence any relationship with the outcomes studied. The available, limited, longitudinal literature suggests that attending certain types of non-parental childcare (particularly informal providers) might be related to less breastfeeding, but the evidence regarding other dietary outcomes is mixed, and sometimes contradictory. Moreover, the data reviewed suggest that attending non-parental childcare is unrelated to physical activity, sedentary behaviour, or sleep outcomes. Included studies were of mixed quality with most (92%) not reporting use of valid and reliable outcome measures, and few (23%) including blinding of outcome assessors to participants’ exposure status. 

### 4.2. Strengths and Limitations of Studies Included in the Review

The measurement of exposure to childcare in included studies was highly variable. Some studies focused on one particular type of childcare provider (e.g., attending Kindergarten Union) [42] versus a reference group that was an amalgam of all other types [40,41,42]. Other studies included only the number of hours per week in non-parental childcare [43,44,45]. Only four studies explored differences between the type of childcare provider [46,49,50,51], but no studies performed detailed analyses exploring differences by duration, intensity, and timing of childcare. Thus, we were unable to explore these as originally intended. 

There was similar heterogeneity in outcome assessment, precluding direct comparisons. Apart from Cairns and Harsh’s study [41], no study reported validity or reliability of the methods used for measuring outcomes. Concurrently, the common use of proxy-report outcome measures increases the risk of measurement error and bias. 

Only seven (54%) studies used an adequate analytical framework that accounted for the potential complexity of the relationship between non-parental childcare and outcomes (physical activity in both cases) [40,42,46,49,50,51], by including and statistically adjusting for potential confounding and mediating variables. Thus, the evidence base may be substantially compromised by uncontrolled confounding by factors such as family and socioeconomic circumstances. Furthermore, apart from Pearce et al.’s [49] study, no other study explored variations in any relationships between exposures and outcomes according to contextual factors, such as socioeconomic circumstances. Thus, we were not able to report on these, as originally planned [34]. Failure to adjust for confounding variables was often a result of the association between childcare and health behaviours not being the primary focus of the study. Greater attention to these associations as primary aims of studies is required to increase the strength of available evidence.

### 4.3. Strengths and Limitations of the Review

This systematic review has several strengths. A large number and variety of databases were searched using a comprehensive search strategy designed with an experienced librarian (V.P.), without limits on date of publication or language. Independent double screening was used at both abstract and full-text screening stages, reducing risk of researcher bias. In cases of remaining uncertainty, study authors were contacted. The risk of bias assessment was also performed independently in duplicate. This review focused on longitudinal studies because these provide a better indication of causality between exposure and outcomes than cross-sectional studies [53].

However, there are a number of limitations that need highlighting. The low number of studies for each of the outcomes did not allow us to present a summary of findings table, nor to perform a meta-analyses as planned [34]. Heterogeneity in the study designs, definition of exposure and outcomes, and the methods and measurement tools used also made comparisons difficult. The results cannot be generalised to middle-income countries, as all studies were located in high-income countries. Furthermore, it was not possible to determine if associations persisted into, or emerged in, adulthood because only two articles and four (4%) associations had outcomes that were measured after age 12 years [42,50].

### 4.4. Interpretation of Findings

Overall, there were substantial null results with a few scattered and mostly inconsistent statistically significant associations between non-parental childcare and diet, physical activity, sedentary behaviour, and sleep outcomes. There was an indication that attending Head Start settings might be associated with positive dietary behaviours compared to other/mixed care (including parental care) [40,47]. However, this evidence comes from only two studies [40,47], and was not seen across all dietary outcomes studied [40], or in relation to all other childcare types. Few consistent findings were found for physical activity, sedentary behaviour or sleep.

Whilst cross-sectional studies generally find more evidence of an unhealthy effect of childcare on diet and activity behaviours [23,24,27,29,31,54,55], this does not appear to be reflected in the limited available longitudinal data. It is possible that any cross-sectional relationship does not persist longitudinally and, hence, that there is no long-term impact of childcare on diet and activity behaviours. This would suggest that identified longitudinal associations between childcare and adiposity occurs via other mechanisms, such as stress. Alternatively, and maybe more likely, the quality and quantity of the longitudinal evidence available on the relationship between childcare and diet and activity behaviours is not strong enough to draw conclusions on the presence or nature of any relationship. 

Most included studies measured outcomes in childhood, up to age 12 years, only [40,41,43,44,45]. It is possible that any effects of childcare on diet and activity behaviours emerge later in life—particularly when children start to develop into more independent adolescents and adults. The significant associations found in D’Onise et al.’s study [42], where physical activity level was assessed during mid-adulthood, support the plausibility of that hypothesis.

The wide range of different outcome and exposure measures used in the included studies indicates poor theorisation and conceptualisation of any potential association. In general, there is limited evidence of shared understanding of exactly what aspect of childcare is expected to be associated with exactly what aspect of diet or type of activity behaviour, what the direction of such associations is, and why. Furthermore, authors rarely addressed the many dimensions that can vary in the exposure to non-parental childcare in terms of provider, timing, duration and intensity (particularly in relation to the activity behaviours outcomes). Greater conceptual clarity in these areas may help drive stronger longitudinal investigations. Clearer disentanglement of all of the potential dimensions in which exposure to childcare may vary will help identify if there are more and less healthful ways in which to provide childcare.

Although it was not possible to perform meta-analyses or meta-regressions, there is no obvious indication that results were related to study size, whether outcomes were considered as continuous and categorical variables, whether outcomes were measured using subjective or objective methods, and whether studies were prospective or retrospective. However, the very small number of studies included for each outcome makes it difficult to draw conclusions on these issues.

### 4.5. Implications for Policy, Practice, and Research

Although overall there was little evidence of a longitudinal relationship between childcare and diet and physical activity, sedentary behaviour, and sleep, this more likely represents an absence of high quality evidence, rather than good evidence of absence of an effect. Given this, it is difficult to draw any firm implications for policy and practice. Nevertheless, and given that there is some evidence of an association between childcare and adiposity [17,18,19], it would be prudent for those regulating and providing childcare to continue to consider how they can provide a healthful environment for children.

The small number of studies included in the current review highlights the need for more longitudinal studies in this area. These studies should employ valid and reliable measures of both exposures and outcomes; analytical frameworks that recognise the potential complexity of the relationships studied, and account for known and possible confounding and mediating factors (e.g., socioeconomic status and maternal employment). Additionally, studies should also perform more detailed investigations to explore potential differences in the effect of childcare according to the type of provider, duration, intensity, and timing of childcare. This would help in clarifying whether specific patterns of exposure to non-parental childcare have a more or less healthful impact on children’s diet or activity behaviours. 

The majority of studies included in this review assessed outcomes up to age 12 years, with only two studies assessing outcomes later in life [42,50]. There is a need for more studies examining long-term relationships, to assess whether relationships between childcare and diet, physical activity, sedentary behaviour, or sleep emerge and continue into adolescence and adulthood. Existent birth cohorts may be useful in this respect.

There is a lack of studies in middle-income countries, as well as consideration of differences in effect by ethnicity and socioeconomic status. Studies in these areas would allow us to assess whether context influences the relationship between childcare and diet and activity behaviours, and hence whether targeted interventions may be justified. 

## 5. Conclusions

This review provides the first systematic summary of studies examining the longitudinal relationship between non-parental childcare and diet, physical activity, sedentary behaviour, and sleep. Results were dominated by null findings with little consistent evidence that non-parental childcare was associated with any of the outcomes of interest. However, the available evidence is limited, highly heterogeneous in the definition and measurement of non-parental childcare, diet and activity behaviours, and lacks an in-depth exploration of different aspects of childcare that may influence this relationship, such as the type, duration, or intensity. Further work is required to clearly conceptualise proposed pathways linking childcare with diet and activity behaviours, and to determine whether, what aspects of, and how much exposure to childcare might impact on these outcomes. This may require wider thinking about the whole system of non-parental childcare and use of systems thinking—increasingly recognised as valuable to public health [56]. This would, in turn, help identify potential targets for interventions, policies, or regulations to help childcare settings provide healthful environments for the children in their care. 

## Figures and Tables

**Figure 1 ijerph-16-04652-f001:**
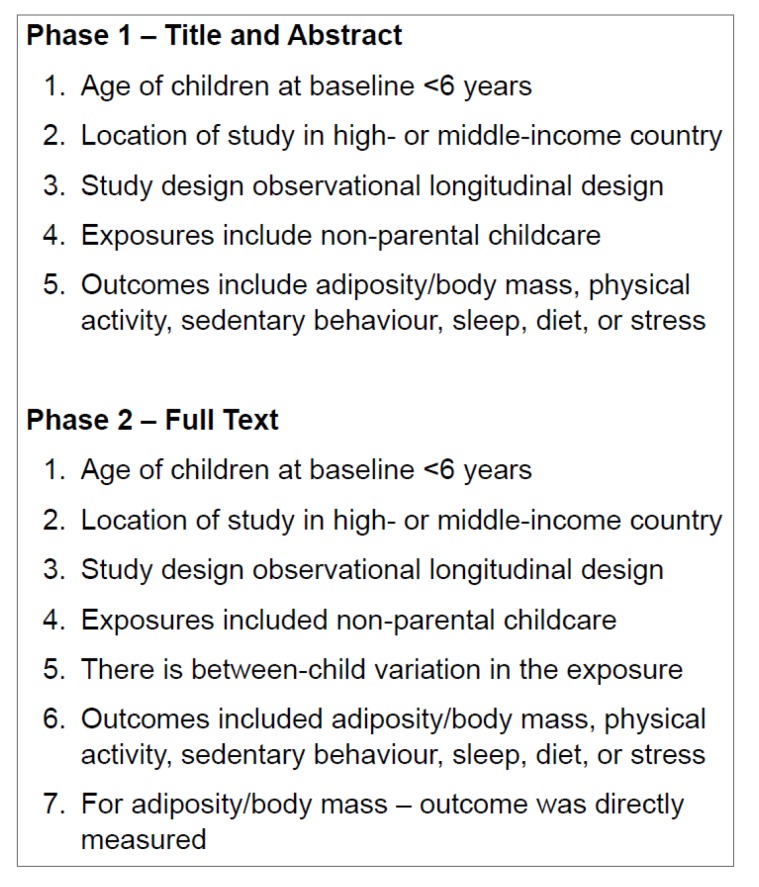
Eligibility criteria.

**Figure 2 ijerph-16-04652-f002:**
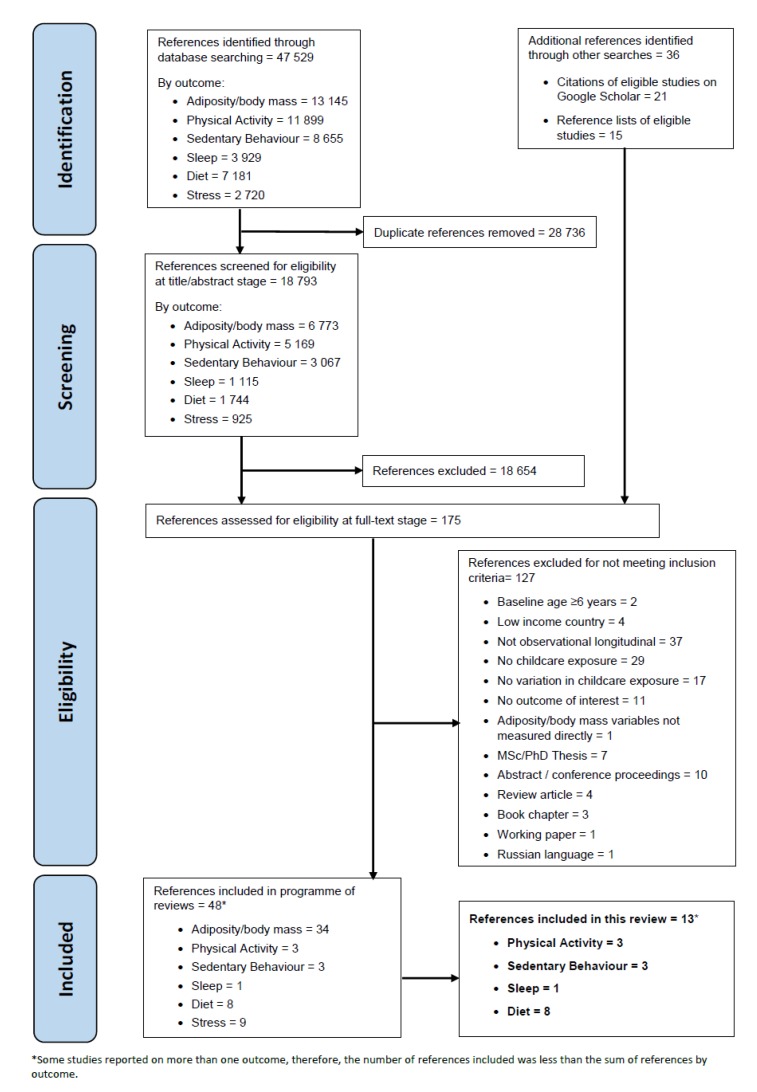
PRISMA diagram.

**Table 1 ijerph-16-04652-t001:** Description of included studies.

Study (Date)	Location	Sample Size	Study Design	Sex	Ethnicity/Country of Birth	Outcome
Belfield & Kelly (2013) [40]	USA	6550	Prospective longitudinal cohort study (Early Childhood Longitudinal Survey—Birth Cohort)	Preschool: 4124 (50.6%) girls, 4026 (49.4%) boys * Kindergarten: 3301 (50.4%) girls, 3249 (49.6%) boys*	Preschool: 1231 (15.1%) Black, 1157 (14.2%) Hispanic, 864 (10.6%) Asian, 1410 (17.3%) Other non-White, 3488 (42.8%) White Kindergarten: 1009 (15.4%) Black, 943 (14.4%) Hispanic, 766 (11.7%) Asian, 1212 (18.5%) Other non-White, 2620 (40.0%) White	Physical activity, diet
Cairns & Harsh (2014) [41]	USA	34	Prospective longitudinal study	15 (44.1%) girls, 19 (55.9%) boys	62% White, 32% Black, 6% Other	Sleep
D’Onise et al. (2011) [42]	Australia	1063	Prospective longitudinal cohort study (North West Adelaide Health Study)	580 (54.6%) girls, 483 (45.4%) boys	Not reported	Physical activity, sedentary behaviour
Lumeng et al. (2005) [43]	USA	1244	Longitudinal panel survey study	630 (50.6%) girls, 614 (49.4%) boys	488 (39.2%) Black, 69 (5.5%) Hispanic, 623 (50.1%) White, 64 (5.1%) Other	Sedentary behaviour
Lumeng et al. (2006) [44]	USA	1016	Prospective longitudinal study	498 (49.0%) girls, 518 (51.0%) boys	841 (82.8%) White, 175 (17.2%) non-White	Sedentary behaviour
Taylor et al. (2009) [45]	New Zealand	3 years: 238 4 years: 216 5 years: 204	Prospective longitudinal cohort study (The Family Lifestyle, Activity, Movement, and Eating study)	3 years: 107 (43.9%) girls, 137 (56.1%) boys 4 years: 104 (43.9%) girls, 133 (56.1%) boys 5 years: 99 (44%) girls, 126 (56%) boys	Baseline: 87% Caucasian, 10.8% Maori, 3.7% Pacific Islanders	Physical activity
Camara et al. (2015) [46]	France	974	Prospective longitudinal cohort study (EDEN—Etude des Déterminants pré et post natals du développement et de la santé de l’ENfant)	454 (46.6%) girls, 520 (53.4%) boys	France birth (ethnic composition not presented)	Diet
Lee et al. (2013) [47]	USA	2150	Prospective longitudinal cohort study (Early Childhood Longitudinal Study-Birth Cohort (ECLS-B))	49% girls, 51% boys (at birth)	USA birth (ethnic composition not presented)	Diet
Levy et al. (2002) [48]	USA	1387	Prospective longitudinal cohort study (Iowa Fluoride Study)	Not reported	95% White, 5% Other	Diet
Pearce et al. (2012) [49]	UK	18,050	Prospective longitudinal cohort study (Millennium Cohort Study)	Not reported	UK birth (ethnic composition not presented)	Diet
Sata et al. (2015) [50]	Japan	4281	Prospective longitudinal cohort study (Ibaraki Children’s Cohort (IBACHIL) Study)	2042 (47.7%) girls, 2239 (52.3%) boys	Japan birth (ethnic composition not presented)	Diet
Wasser et al. (2013) [51]	USA	210	Prospective longitudinal study	116 (53.5%) girls, 101 (46.5%) boys	African-American	Diet
Weile et al. (1990) [52]	Denmark	500	Prospective longitudinal study	Not reported	Denmark birth (ethnic composition not presented)	Diet

Legend: BMI—body mass index; SD—standard deviation. * Numbers are approximate, calculated from the percentages and total preschool and kindergarten sample sizes presented in Belfield and Kelly (2013) Appendix Table A2.

**Table 2 ijerph-16-04652-t002:** Results of included studies.

Study	Exposure	Age at Childcare Exposure	Outcome	Age at Outcome	Analysis	Adjustment	Results (Most Adjusted Model)
Belfield & Kelly (2013)	Centre-based preschool	4 y	Low activity level	5–6 y	Multivariable probit regression models	Child’s race/ethnicity, gender, age, and number of siblings, twin (yes/no), maternal employment, education, and marital status, health insurance status, father non-resident, household income, geographic region, and prior health at 24 mo (general health status, and indicators of asthma, gastroenteritis, respiratory condition, and ear infection)	AME = 0.124 (SE: 0.120), *p* > 0.05
	Centre-based preschool	4 y	Screened low activity	5–6 y	Multivariable probit regression models		AME = 0.056 (SE: 0.084), *p* > 0.05
	Centre-based preschool	4 y	Regular fast food consumption (vs. not)	5–6 y	Multivariable probit regression models		AME = −0.15 (SE: 0.055), *p* > 0.05
	Centre-based preschool	4 y	Regular soda consumption (vs. not)	5–6 y	Multivariable probit regression models		AME = −0.117 (SE: 0.056), *p* < 0.05
	Centre-based preschool	4 y	Regular candy consumption (vs. not)	5–6 y	Multivariable probit regression models		AME = 0.031 (SE: 0.053), *p* > 0.05
	Centre-based preschool	4 y	Regular chips consumption (vs. not)	5–6 y	Multivariable probit regression models		AME = 0.113 (SE: 0.054), *p* < 0.05
	Centre-based preschool	4 y	Infrequent vegetable consumption (vs. frequent)	5–6 y	Multivariable probit regression models		AME = 0.049 (SE: 0.058), *p* > 0.05
	Centre-based preschool	4 y	Infrequent fruit consumption (vs. frequent)	5–6 y	Multivariable probit regression models		AME = −0.120 (SE: 0.060), *p* < 0.05
	Centre-based preschool for 2 y	4 y	Low activity level	5–6 y	Multivariable probit regression models		AME = 0.007 (SE: 0.173), *p* > 0.05
	Centre-based preschool for 2 y	4 y	Screened low activity	5–6 y	Multivariable probit regression models		AME = 0.058 (SE: 0.126), *p* > 0.05
	Centre-based preschool for 2 y	4 y	Regular fast food consumption (vs. not)	5–6 y	Multivariable probit regression models		AME = 0.022 (SE: 0.085), *p* > 0.05
	Centre-based preschool for 2 y	4 y	Regular soda consumption (vs. not)	5–6 y	Multivariable probit regression models		AME = −0.260 (SE: 0.083), *p* < 0.01
	Centre-based preschool for 2 y	4 y	Regular candy consumption (vs. not)	5–6 y	Multivariable probit regression models		AME = −0.031 (SE: 0.080), *p* > 0.05
	Centre-based preschool for 2 y	4 y	Regular chips consumption (vs. not)	5–6 y	Multivariable probit regression models		AME = 0.024 (SE: 0.081), *p* > 0.05
	Centre-based preschool for 2 y	4 y	Infrequent vegetable consumption (vs. frequent)	5–6 y	Multivariable probit regression models		AME = −0.113 (SE: 0.090), *p* > 0.05
	Centre-based preschool for 2 y	4 y	Infrequent fruit consumption (vs. frequent)	5–6 y	Multivariable probit regression models		AME = −0.231 (SE: 0.093), *p* < 0.01
	Head Start	4 y	Low activity level	5–6 y	Multivariable probit regression models		AME = 0.313 (SE: 0.142), *p* < 0.05
	Head Start	4 y	Screened low activity	5–6 y	Multivariable probit regression models		AME = 0.128 (SE: 0.112), *p* > 0.05
	Head Start	4 y	Regular fast food consumption (vs. not)	5–6 y	Multivariable probit regression models		AME = 0.050 (SE: 0.077), *p* > 0.05
	Head Start	4 y	Regular soda consumption (vs. not)	5–6 y	Multivariable probit regression models		AME = −0.065 (SE: 0.081), *p* > 0.05
	Head Start	4 y	Regular candy consumption (vs. not)	5–6 y	Multivariable probit regression models		AME = −0.108 (SE: 0.073), *p* > 0.05
	Head Start	4 y	Regular chips consumption (vs. not)	5–6 y	Multivariable probit regression models		AME = 0.040 (SE: 0.074), *p* > 0.05
	Head Start	4 y	Infrequent vegetable consumption (vs. frequent)	5–6 y	Multivariable probit regression models		AME = −0.067 (SE: 0.083), *p* > 0.05
	Head Start	4 y	Infrequent fruit consumption (vs. frequent)	5–6 y	Multivariable probit regression models		AME = −0.266 (SE: 0.085), *p* < 0.01
Cairns & Harsh (2014)	All day preschool/daycare (vs. primary/secondary caregiver)	5 y ^a^	Total sleep duration weekday	5 y ^b^	Group by assessment mixed model ANOVA	None reported	Group: Not significant Time: F(2,64) = 5.2, *p* = 0.008, η^2^ = 0.14 Group by time: Not significant
	All day preschool/daycare (vs. primary/secondary caregiver)	5 y ^a^	Nocturnal sleep duration weekday	5 y ^b^	Group by assessment mixed model ANOVA		Group: Not significant Time: Not significant Group by time: Not significant
	All day preschool/daycare (vs. primary/secondary caregiver)	5 y ^a^	Sleep onset weekday	5 y ^b^	Group by assessment mixed model ANOVA		Group: F(1,32) = 5.8, *p* = 0.022, η^2^ = 0.15 Time: F(2,64) = 40.9, *p* < 0.001, η^2^ = 0.56 Group by time: F(2,64) = 6.1, *p* = 0.004, η^2^ = 0.16
	All day preschool/daycare (vs. primary/secondary caregiver)	5 y ^a^	Sleep onset time weekend	5 y ^b^	Group by assessment mixed model ANOVA		Group: Not reported Time: F(2,64) = 6.9, *p* < 0.01, η^2^ = 0.18 Group by time: Not reported
	All day preschool/daycare (vs. primary/secondary caregiver)	5 y ^a^	Wake up time weekday	5 y ^b^	Group by assessment mixed model ANOVA		Group: F(1,32) = 14.9, *p* = 0.001, η^2^ = 0.32 Time: F(2,64) = 81.6, *p* < 0.001, η^2^ = 0.72 Group by time: F(2,64) = 17.5, *p* < 0.001, η^2^ = 0.35
	All day preschool/daycare (vs. primary/secondary caregiver)	5 y ^a^	Wake up time weekend	5 y ^b^	Group by assessment mixed model ANOVA		Group: Not reported Time: F(2,64) = 4.4, *p* < 0.05, η^2^ = 0.12 Group by time: Not reported
	All day preschool/daycare (vs. primary/secondary caregiver)	5 y ^a^	Sleep efficiency weekday	5 y ^b^	Group by assessment mixed model ANOVA		Group: Not significant Time: F(2,64) = 3.5, *p* < 0.05, η^2^ = 0.10 Group by time: Not significant
	All day preschool/daycare (vs. primary/secondary caregiver)	5 y ^a^	Nap duration weekday	5 y ^b^	Group by assessment mixed model ANOVA		Group: Not reported Time: F(2,55) = 20.46, *p* < 0.001, η^2^ = 0.436 Group by time: Not reported
	All day preschool/daycare (vs. primary/secondary caregiver)	5 y ^a^	Nap duration weekend	5 y ^b^	Group by assessment mixed model ANOVA		Group: Not reported Time: Not significant Group by time: Not reported
	All day preschool/daycare (vs. primary/secondary caregiver)	5 y ^a^	Number of weekdays with a nap	5 y ^b^	T-test (Summer vs. 2 weeks after start of kindergarten)		Group: Not reported Time: T(13) = 3.4, *p* = 0.005 Group by time: Not reported
	All day preschool/daycare (vs. primary/secondary caregiver)	5 y ^a^	Number of weekday naps	5 y ^b^	Group by assessment mixed model ANOVA		Group: Not significant Time: Not significant Group by time: Not reported
	All day preschool/daycare (vs. primary/secondary caregiver)	5 y ^a^	Number of weekend naps	5 y ^b^	Group by assessment mixed model ANOVA		Group: Not significant Time: Not significant Group by time: Not reported
	All day preschool/daycare (vs. primary/secondary caregiver)	5 y ^a^	Caregivers rating children as having less difficulty in going to bed	5 y ^b^	Group by assessment mixed model ANOVA		Group: Not reported Time: F(2,55) = 20.46, *p* < 0.001, η^2^ = 0.436 Group by time: Not reported
	All day preschool/daycare (vs. primary/secondary caregiver)	5 y ^a^	Caregivers rating children as having less difficulty falling asleep	5 y ^b^	Group by assessment mixed model ANOVA		Group: Not reported Time: F(2,42) = 3.9, *p* = 0.03, η^2^ = 0.16 Group by time: Not reported
	All day preschool/daycare (vs. primary/secondary caregiver)	5 y ^a^	Caregivers ratings of returning to wakefulness in the morning	5 y ^b^	Group by assessment mixed model ANOVA		Group: Not reported Time: Not reported Group by time: F(2,42) = 6.3, *p* = 0.004, η^2^ = 0.23
Camara et al. (2015)	Childcare arrangement	2–3 y	Processed, fast-foods at 2, 3, and 5 y dietary pattern	2, 3, 5 y	Multivariable linear regression	Child’s age, gender, recruitment centre, season when the food frequency questionnaire was completed household disadvantage composite index, older sibling at home (2 y), maternal age at delivery, education level, and current/ past occupation, working time, and unemployed/student when child aged 2 y	At home, cared for by mother: Reference At home, not cared for by mother: B = 0.70 (SE: 0.14), *p* < 0.001 Crèche/pre-school: B = −0.03 (SE: 0.13), *p* > 0.05 At nanny’s home: B = 0.13 (SE: 0.13), *p* > 0.05
	Childcare arrangement	2–3 y	Guidelines at 2, 3 and 5 y dietary pattern	2, 3, 5 y	Multivariable linear regression		At home, cared for by mother: Reference At home, not cared for by mother: B = 0.01 (SE:0.15), *p* > 0.05 Crèche/pre-school: B = 0.08 (SE: 0.13), *p* > 0.05 At nanny’s home: B = 0.10 (SE: 0.13), *p* > 0.05
D’Onise et al. (2011)	Attended Kindergarten Union preschool (vs. not attended)	2–5 y	PA level	Preschool mean: 45.3 y No Preschool mean: 51.1 y	Multinomial logistic regression	Age, gender, child socioeconomic position, adult height, educational attainment, and adult income	Sedentary: Reference Low PA: RRR = 1.24 (95%CI: 0.89-1.74) Moderate PA: RRR = 1.26 (95%CI: 0.87–1.81) High PA: RRR = 1.99 (95%CI: 1.19–3.35)
Lee et al. (2013)	Type of childcare arrangement on a regular basis—Head Start vs. not Head Start	4 y	Frequency of having healthy eating patterns (times/week)	5–6 y	Propensity score-weighted regressions	Child’s variables: (e.g., gender, ethnicity, multiple birth, prematurity, breastfeeding and number of siblings at 9 mo, pre-treatment outcomes at 2 y); Maternal variables: (e.g., married at birth (yes/no), pre-pregnancy age and body mass index, depression at 9 mo, health status and employment at 2 y, foreign born); parenting behaviours/ home environments (e.g., KIDI at 9 mo, having sleep routine, weekday watching TV, and indoor and outdoor activities at 2 y); Family variables: (e.g., parent’s education at birth, parental occupation and family income at 9 mo, living in urban area, region of country, and number of times receiving Special Supplemental Nutrition Program for Women, Infants, and Children, food stamps, and Temporary Assistance for Needy Families by 2 y)	M = 2.21 (SE: 0.74), *p* < 0.01
	Type of childcare arrangement on a regular basis—Head Start vs. not Head Start	4 y	Frequency of having unhealthy eating patterns (times/week)	5–6 y	Propensity score-weighted regressions		M = 0.63 (SE: 0.57), *p* < 0.05
	Type of childcare arrangement on a regular basis—Head Start vs. Pre-Kindergarten	4 y	Frequency of having healthy eating patterns (times/week)	5–6 y	Propensity score-weighted regressions		M = 1.26 (SE: 1.33), *p* > 0.05
	Type of childcare arrangement on a regular basis—Head Start vs. Pre-Kindergarten	4 y	Frequency of having unhealthy eating patterns (times/week)	5–6 y	Propensity score-weighted regressions		M = 0.36 (SE: 0.97), *p* < 0.05
	Type of childcare arrangement on a regular basis—Head Start vs. Other centre-based	4 y	Frequency of having healthy eating patterns (times/week)	5–6 y	Propensity score-weighted regressions		M = 2.35 (SE: 1.14), *p* < 0.05
	Type of childcare arrangement on a regular basis—Head Start vs. Other centre-based	4 y	Frequency of having unhealthy eating patterns (times/week)	5–6 y	Propensity score-weighted regressions		M = 0.80 (SE: 0.78), *p* < 0.05
	Type of childcare arrangement on a regular basis—Head Start vs. Other non-parental	4 y	Frequency of having healthy eating patterns (times/week)	5–6 y	Propensity score-weighted regressions		M = 2.74 (SE: 1.32), *p* < 0.05
	Type of childcare arrangement on a regular basis—Head Start vs. Other non-parental	4 y	Frequency of having unhealthy eating patterns (times/week)	5–6 y	Propensity score-weighted regressions		M = 0.77 (SE:0.98), *p* < 0.05
	Type of childcare arrangement on a regular basis—Head Start vs. parental	4 y	Frequency of having healthy eating patterns (times/week)	5–6 y	Propensity score-weighted regressions		M = 2.07 (SE: 1.01), *p* < 0.05
	Type of childcare arrangement on a regular basis—Head Start vs. parental	4 y	Frequency of having unhealthy eating patterns (times/week)	5–6 y	Propensity score-weighted regressions		M = 0.47 (SE: 0.77), *p* < 0.05
Levy et al. (2002)	Number of days in childcare between 0–6 mo of age	6 weeks, 3 months, 6 months (referring to preceding time period)	Time until cessation of breastfeeding	6 weeks, 3 months, 6 months	Cox proportional hazard regression	Pacifier use, digit sucking, maternal and paternal age and education, family income, breastfeeding plans, maternal smoking, infant’s gender, and infant antibiotic use.	No pacifier use, or digit sucking, or childcare: Reference No pacifier use, does digit sucking, no childcare days: *p* ≥ 0.05 No pacifier use, does digit sucking, 15 childcare days: HR = 1.41 (95%CI: 1.02–1.96), *p* < 0. No pacifier use, does digit sucking, 30 childcare days: *p* ≥ 0.05 No pacifier use, does digit sucking, 60 childcare days: *p* ≥ 0.05 Pacifier use, no digit sucking, no childcare: HR = 1.67 (95%CI: Not reported #), *p* < 0.05 Pacifier use, no digit sucking, 15 days childcare: *p* ≥ 0.05 Pacifier use, no digit sucking, 30 days childcare: Significant # Pacifier use, no digit sucking, 60 days childcare: Borderline significant # Pacifier use and digit sucking, no childcare: HR = 1.88 (95%CI: 1.36–2.62), *p* < 0.05 Pacifier use and digit sucking, 15 childcare days: HR = 1.52 (95%CI: 1.03–2.25), *p* < 0.05 Pacifier use and digit sucking, 30 childcare days: *p* ≥ 0.05 Pacifier use and digit sucking, 60 childcare days: Not significant
Lumeng et al. (2005)	Centre-based childcare attendance intensity (none vs. 15 h/week vs. ≥15 h/week)	3–5 y	>4 h/day of TV viewing (yes vs. no)	6–12 y	Turkey’s test	None	Not significant, *p* = 0.27
Lumeng et al. (2006)	Average number of hours in non-parental childcare	24–36 months	TV viewing (<2 h/day vs. ≥2 h/day)	36 months	*t*-test	None	Not significant, *p* = 0.58
Pearce et al. (2012)	Overall childcare type	<4 to 9 months	Breastfeeding for ≥4 months	9 months	Poisson regression	Mother’s ethnicity, parity, age at first live birth, and whether the mother returned to work before the infant was age 4 mo	Parent: Reference Informal: RR = 0.51 (95%CI: 0.43–0.59), *p* < 0.05 Formal: RR = 0.84 (95%CI: 0.72–0.99), *p* < 0.05
	Childcare type by intensity	<4 to 9 months	Breastfeeding for ≥4 months	9 months	Poisson regression		Parent: Reference Informal part-time: RR = 0.54 (95%CI: 0.45–0.63), *p* < 0.05 Informal full-time: RR = 0.42 (95%CI: 0.28–0.64), *p* < 0.05 Formal part-time: RR = 1.01 (95%CI: 0.82–1.24), *p*≥0.05 Formal full-time: RR = 0.68 (95%CI: 0.51–0.92), *p* < 0.05
	Childcare type by National Statistics Socio-economic Classification	<4 to 9 months	Breastfeeding for ≥4 months	9 months	Poisson regression		Routine and Manual: Parent: Reference Informal: RR = 0.47 (95%CI: 0.34–0.66), *p* < 0.05 Formal: RR = 0.54 (95%CI: 0.21–1.36), *p* ≥ 0.05 Intermediate: Parent: Reference Informal: RR = 0.50 (95%CI: 0.37–0.67), *p* < 0.05 Formal: RR = 0.84 (95%CI: 0.57–1.23), *p* ≥ 0.05 Managerial and Professional: Parent: Reference Informal: RR = 0.50 (95%CI: 0.39–0.65), *p* < 0.05 Formal: RR = 0.76 (95%CI: 0.62–0.94), *p* < 0.05
	Childcare type by Maternal Education	<4 to 9 months	Breastfeeding for ≥4 months	9 months	Poisson regression		None–GCSE D–G: Parent: Reference Informal: RR = 0.44 (95%CI: 0.27–0.71), *p* < 0.05 Formal: RR = 1.00 (95%CI: 0.44–2.28), *p* ≥ 0.05 GCSE A–C, A Levels, Diploma: Parent: Reference Informal: RR = 0.47 (95% CI: 0.37–0.59), *p* ≥ 0.05 Formal: RR = 0.83 (95%CI: 0.64–1.08), *p* ≥ 0.05 Degree: Parent: Reference Informal: RR = 0.82 (95%CI: 0.64–1.06), *p* ≥ 0.05 Formal: RR = 0.71 (95%CI: 0.58–0.86), *p* < 0.05
	Childcare type by Lone Parenthood	<4 to 9 months	Breastfeeding for ≥4 months	9 months	Poisson regression		Lone parent: Parent: Reference Informal: RR = 0.40 (95%CI: 0.25–0.65), *p* < 0.05 Formal: RR = 1.65 (95%CI: 1.04–2.63), *p* < 0.05 Couple family: Parent: Reference Informal: RR = 0.53 (95%CI: 0.44–0.63), *p* < 0.05 Formal: RR = 0.79 (95%CI: 0.66–0.94), *p* < 0.05
	Childcare type by Area Deprivation	<4 to 9 months	Breastfeeding for ≥4 months	9 months	Poisson regression		5 (Most deprived): Parent: Reference Informal: RR = 0.72 (95%CI: 0.53–0.97), *p* < 0.05 Formal: RR = 0.63 (95% CI: 0.28–1.39), *p* ≥ 0.05 4: Parent: Reference Informal: RR = 0.54 (95%CI: 0.36–0.81), *p* < 0.05 Formal: RR = 1.12 (95% CI: 0.73–1.72), *p* ≥ 0.05 3: Parent: Reference Informal: RR = 0.51 (95%CI: 0.33–0.80), *p* < 0.05 Formal: RR = 1.27 (95% CI: 0.86–1.85), *p* ≥ 0.05 2: Parent: Reference Informal: RR = 0.37 (95%CI: 0.21–0.65), *p* < 0.05 Formal: RR = 0.71 (95% CI: 0.47–1.06), *p* ≥ 0.05 1 (Least deprived) Parent: Reference Informal: RR = 0.48 (95%CI: 0.26–0.88), *p* < 0.05 Formal: RR = 0.64 (95% CI: 0.42–1.00), *p* ≥ 0.05
Sata et al. (2015)	Main daytime caregiver	3 y	Between-meal eating before dinner	6 y	Logistic regression models, stratified by gender	Baseline types of feeding, wake-up time, time of sleep, physical activity, playing outside, living with brothers or sisters, picky eating, and father’s employment.	Boys: Mothers: Reference Grandparents: OR = 2.1 (95%CI: 1.4–3.1), *p* < 0.001 Nursery school/kindergarten staff: OR = 1.6 (95%CI: 1.1–2.4), *p* < 0.05 Girls: Mothers: Reference Grandparents: OR = 2.5 (95%CI: 1.7–3.8), *p* < 0.001 Nursery school/kindergarten staff: OR = 1.6 (95%CI: 1.1–2.4), *p* < 0.05
	Main daytime caregiver	3 y	Between-meal eating ≥3 times/day	6 y	Logistic regression models, stratified by gender		Boys: Mothers: Reference Grandparents: OR = 3.2 (95%CI: 1.3–7.7), *p* < 0.05 Nursery school/kindergarten staff: OR = 1.9 (95%CI: 0.7–5.4), *p* ≥ 0.05 Girls: Mothers: Reference Grandparents: OR = 2.7 (95%CI: 1.1–6.7), *p* < 0.05 Nursery school/kindergarten staff: OR = 2.3 (95%CI: 0.9–6.3), *p* ≥ 0.05
	Main daytime caregiver	3 y	Between-meal eating before bedtime ≥3 times/week	6 y	Logistic regression models, stratified by gender		Boys: Mothers: Reference Grandparents: OR = 1.5 (95%CI: 0.8–2.7), *p* ≥ 0.05 Nursery school/kindergarten staff: OR = 1.1 (95% CI: 0.6–2.0), *p* ≥ 0.05 Girls: Mothers: Reference Grandparents: OR = 1.4 (95%CI: 0.7–2.5), *p* ≥ 0.05 Nursery school/kindergarten staff: OR = 1.6 (95% CI: 0.8–3.0), *p* ≥ 0.05
	Main daytime caregiver	3 y	Between-meal eating before dinner	12 y	Logistic regression models, stratified by gender		Boys: Mothers: Reference Grandparents: OR = 1.3 (95%CI: 0.9–1.8), *p* ≥ 0.05 Nursery school/kindergarten staff: OR = 1.0 (95%CI: 0.7–1.5), *p* ≥ 0.05 Girls: Mothers: Reference Grandparents: OR = 1.9 (95%CI: 1.3–2.8), *p* < 0.01 Nursery school/kindergarten staff: OR = 1.7 (95%CI: 1.1–2.5), *p* < 0.05
	Main daytime caregiver	3 y	Between-meal eating ≥5 times/week	12 y	Logistic regression models, stratified by gender		Boys: Mothers: Reference Grandparents: OR = 1.0 (95%CI: 0.7–1.4), *p* ≥ 0.05 Nursery school/kindergarten staff: OR = 1.2 (95%CI: 0.8–1.7), *p* ≥ 0.05 Girls: Mothers: Reference Grandparents: OR = 0.9 (95%CI: 0.6–1.3), *p* ≥ 0.05 Nursery school/kindergarten staff: OR = 0.9 (95%CI: 0.6–1.3), *p* ≥ 0.05
	Main daytime caregiver	3 y	Between-meal eating before bedtime ≥3 times/week	12 y	Logistic regression models, stratified by gender		Boys: Mothers: Reference Grandparents: OR = 1.5 (95%CI: 0.9–2.5), *p* ≥ 0.05 Nursery school/kindergarten staff: OR = 0.7 (95%CI: 0.4–1.3), *p* ≥ 0.05 Girls: Mothers: Reference Grandparents: OR = 1.1 (95%CI: 0.6–2.2), *p* ≥ 0.05 Nursery school/kindergarten staff: OR = 1.2 (95%CI: 0.6–2.5), *p* ≥ 0.05
	Main daytime caregiver	3 y	Between-meal eating before dinner	22 y	Logistic regression models, stratified by gender		Boys: Mothers: Reference Grandparents: OR = 0.9 (95%CI: 0.6–1.5), *p* ≥ 0.05 Nursery school/kindergarten staff: OR = 1.2 (95%CI: 0.8–1.9), *p* ≥ 0.05 Girls: Mothers: Reference Grandparents: OR = 1.2 (95%CI: 0.7–2.0), *p* ≥ 0.05 Nursery school/kindergarten staff: OR = 0.9 (95%CI: 0.5–1.5), *p* ≥ 0.05
	Main daytime caregiver	3 y	Between-meal eating ≥5 times/week	22 y	Logistic regression models, stratified by gender		Boys: Mothers: Reference Grandparents: OR = 0.9 (95%CI: 0.5–1.5) Nursery school/kindergarten staff: OR = 1.0 (95%CI: 0.6–1.6) Girls: Mothers: Reference Grandparents: OR = 0.8 (95%CI: 0.5–1.3) Nursery school/kindergarten staff: OR = 1.1 (95%CI: 0.7–1.8)
	Main daytime caregiver	3 y	Between-meal eating before bedtime ≥3 times/week	22 y	Logistic regression models, stratified by gender		Boys: Mothers: Reference Grandparents: OR = 1.0 (95%CI: 0.6–1.7), *p* ≥ 0.05 Nursery school/kindergarten staff: OR = 0.6 (95%CI: 0.3–1.1), *p* ≥ 0.05 Girls: Mothers: Reference Grandparents: OR = 1.3 (95%CI: 0.6–2.6), *p* ≥ 0.05 Nursery school/kindergarten staff: OR = 1.3 (95%CI: 0.6–2.8), *p* ≥ 0.05
Taylor et al. (2009)	Number of hours per week childcare attendance	3, 4, 5 y	Total active time (minutes/day)	3, 4, 5 y	Random coefficient regression	None reported	Not significant, *p* = 0.069–0.806
	Number of hours per week childcare attendance	3, 4, 5 y	Average accelerometer counts (counts/minute)	3, 4, 5 y	Random coefficient regression	None reported	Not significant, *p* = 0.069–0.806
Wasser et al. (2013)	Any non-maternal caregiver use	6–18 months	Consuming any whole fruit	6–18 months	Random-effects logistic regression	Maternal age, employment, depression, any maternal college, and marital status	None: Reference Any: OR = 1.51 (95%CI: 1.03–2.23), *p* < 0.05
	Any non-maternal caregiver use	6–18 months	Consuming any vegetable	6–18 months	Random-effects logistic regression		None: Reference Any: OR = 1.25 (95%CI: 0.79–1.99), *p* ≥ 0.05
	Any non-maternal caregiver use	6–18 months	Consuming any juice	6–18 months	Random-effects logistic regression		None: Reference Any: OR = 1.64 (95%CI: 1.01–2.67), *p* < 0.05
	Any non-maternal caregiver use	6–18 months	Consuming any fried potatoes	6–18 months	Random-effects logistic regression		None: Reference Any: OR = 0.82 (95%CI: 0.46–1.43), *p* ≥ 0.05
	Any non-maternal caregiver use	6–18 months	Consuming any desserts and sweets	6–18 months	Random-effects logistic regression		None: Reference Any: OR = 1.20 (95%CI: 0.77–1.86), *p* ≥ 0.05
	Any non-maternal caregiver use	6–18 months	Consuming any sweetened beverages	6–18 months	Random-effects logistic regression		None: Reference Any: OR = 1.17 (95%CI: 0.65–2.12), *p* ≥ 0.05
	Any non-maternal caregiver use	6–18 months	Consuming any salty snacks	6–18 months	Random-effects logistic regression		None: Reference Any: OR = 1.45 (95%CI: 0.67–3.12), *p* ≥ 0.05
	Type of non-maternal caregiver use	6–18 months	Consuming any whole fruit	6–18 months	Random-effects logistic regression		None: Reference Father: OR = 1.12 (95%CI: 0.64–1.97), *p* ≥ 0.05 Grandmother: OR = 0.92 (95%CI: 0.57–1.5), *p* ≥ 0.05 Licensed provider: OR = 1.55 (95%CI: 0.93–2.59), *p* ≥ 0.05
	Type of non-maternal caregiver use	6–18 months	Consuming any vegetable	6–18 months	Random-effects logistic regression		None: Reference Father: OR = 0.93 (95%CI: 0.48–1.8), *p* ≥ 0.05 Grandmother: OR = 0.89 (95%CI: 0.5–1.59), *p* ≥ 0.05 Licensed provider: OR = 0.96 (95%CI: 0.52–1.79), *p* ≥ 0.05
	Type of non-maternal caregiver use	6–18 months	Consuming any juice	6–18 months	Random-effects logistic regression		None: Reference Father: OR = 0.83 (95%CI: 0.42–1.64), *p* ≥ 0.05 Grandmother: OR = 1.97 (95%CI: 1.02–3.81), *p* < 0.05 Licensed provider: OR = 1.2 (95%CI: 0.61–2.34), *p* ≥ 0.05
	Type of non-maternal caregiver use	6–18 months	Consuming any fried potatoes	6–18 months	Random-effects logistic regression		None: Reference Father: OR = 1.13 (95%CI: 0.48–2.69), *p* ≥ 0.05 Grandmother: OR = 0.97 (95%CI: 0.48–1.96), *p* ≥ 0.05 Licensed provider: OR = 0.75 (95%CI: 0.38–1.48), *p* ≥ 0.05
	Type of non-maternal caregiver use	6–18 months	Consuming any desserts and sweets	6–18 months	Random-effects logistic regression		None: Reference Father: OR = 0.85 (95%CI: 0.44–1.67), *p* ≥ 0.05 Grandmother: OR = 0.74 (95%CI: 0.42–1.28), *p* ≥ 0.05 Licensed provider: OR = 1.30 (95%CI: 0.75–2.26), *p* ≥ 0.05
	Type of non-maternal caregiver use	6–18 months	Consuming any sweetened beverages	6–18 months	Random-effects logistic regression		None: Reference Father: OR = 1.71 (95%CI: 0.71–4.11), *p* ≥ 0.05 Grandmother: OR = 0.97 (95%CI: 0.46–2.05), *p* ≥ 0.05 Licensed provider: OR = 1.28 (95%CI: 0.63–2.62), *p*≥0.05
	Type of non-maternal caregiver use	6–18 months	Consuming any salty snacks	6–18 months	Random-effects logistic regression		None: Reference Father: OR = 2.06 (95%CI: 0.66-6.39), *p* ≥ 0.05 Grandmother: OR = 1.03 (95%CI: 0.40–2.69), *p* ≥ 0.05 Licensed provider: OR = 0.71 (95%CI: 0.28–1.79), *p* ≥ 0.05
Weile et al. (1990)	Attending daycare (vs. cared for at home)	1–12 months	Changing from feeding categories 1/2 to categories 3/4/5 *	1–12 months	Cox proportional hazards model	Other children in family and socioeconomic status	RR = 2.08 (95%CI: 1.43–3.01), *p* < 0.05
	Attending daycare (vs. cared for at home)	1–12 months	Changing from feeding categories 1/2/3 to categories 4/5 *	1–12 months	Cox proportional hazards model	Other children in family and socioeconomic status	RR = 2.05 (95%CI:1.39–3.02), *p* < 0.05
	Attending daycare (vs. cared for at home)	1–12 months	Changing from feeding categories 1/2/3/4 to category 5 *	1–12 months	Cox proportional hazards model	Other children in family and socioeconomic status	RR = 2.50 (95%CI: 1.66–3.78), *p* < 0.05

Legend: AME—average marginal effects; CI—confidence interval; h—hours; mo—months; y—years; OR—odds ratio; PA—physical activity; RR—relative risk; RRR—relative risk ratio; SD—standard deviation; SE—robust standard errors; TV—television; y—years. ^a^ 2–3 weeks before start of kindergarten; ^b^ 2 weeks, 1 month after start of kindergarten; * Categories: (1) 100% breast-fed, (2) breast-fed > formula-fed, (3) breast-fed = formula-fed, (4) breast-fed < formula-fed, and (5) 100% formula-fed. # Estimates and significance figures taken from text only, as article did not present tables and we could not obtain these from publishers or authors.

**Table 3 ijerph-16-04652-t003:** Results of the Nutrition Evidence Library Bias Assessment Tool risk of bias evaluation.

NEL-BAT Question	Belfield & Kelly (2013) [40]	Cairns & Harsh (2014) [41]	D’Onise et al. (2011) [42]	Lumeng et al. (2005) [43]	Lumeng et al. (2006) [44]	Taylor et al. (2009) [45]	Camara et al. (2015) [46]	Lee et al. (2013) [47]	Levy et al. (2002) [48]	Pearce et al. (2012) [49]	Stata et al. (2015) [50]	Wasser et al. (2013) [51]	Weile et al. (1990) [52]	Total Score by Question
1.Were the inclusion/exclusion criteria similar across study groups?	0	0	0	0	0	0	0	0	0	0	0	0	0	0
2. Was the strategy for recruiting or allocating participants similar across study groups?	0	0	0	0	0	0	0	0	0	0	0	0	0	0
5. Was there an attempt to balance the allocation between the study groups or match the study groups (e.g., through stratification, matching, propensity scores)?	0	2	0	2	0	0	1	0	1	1	2	1	0	10
6. Was distribution of health status, demographics, and other critical confounding factors similar across study groups at baseline? If not, does the analysis control for baseline differences between groups?	0	1	0	2	2	0	0	0	1	1	2	1	1	11
7. Did the investigators account for important variations in the execution of the study from the proposed protocol or research plan?	0	0	0	1	0	1	0	0	0	0	0	0	0	2
8. Was adherence to the study protocol similar across study groups?	0	0	0	0	0	0	1	1	1	1	1	1	1	7
9. Did the investigators account for the impact of unintended/unplanned concurrent interventions or exposures that were differentially experienced by study groups and might bias results?	0	0	0	0	0	0	1	1	1	1	1	1	0	6
12. Were outcome assessors blinded to the intervention or exposure status of participants?	0	1	1	0	0	1	1	2	2	1	2	1	2	14
13. Were valid and reliable measures used consistently across all study groups to assess inclusion/exclusion criteria, interventions/exposures, outcomes, participant health benefits and harms, and confounding?	1	0	2	2	2	2	2	2	2	1	2	2	2	22
14. Was the length of follow-up similar across study groups?	0	0	2	0	0	0	0	0	0	0	0	0	0	2
15. In cases of high or differential loss to follow-up, was the impact assessed (e.g., through sensitivity analysis or other adjustment method)?	0	1	1	1	1	0	2	0	1	0	2	0	0	9
16. Were other sources of bias taken into account in the design and/or analysis of the study (e.g., through matching, stratification, interaction terms, multivariate analysis, or other statistical adjustment such as instrumental variables)?	0	2	0	2	2	2	0	0	0	0	0	0	0	8
17.Were the statistical methods used to assess the primary outcomes adequate?	0	0	0	2	2	2	0	0	0	0	0	0	0	6
**Total study score:**	1	7	6	12	9	8	8	6	9	6	12	7	6

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
