# Peer review of "Relationship Between Early Childhood Non-Parental Childcare and Diet, Physical Activity, Sedentary Behaviour, and Sleep: A Systematic Review of Longitudinal Studies"

_ijerph, 2019, doi:10.3390/ijerph16234652_

Round 1
Reviewer 1 Report
This an exemplar of a systematic review; well organized and clear.
Minor suggestions;
Line 42: clarify the ecological approach used with perhaps a reference to Bronfenbrenner and include some indication of the complexity of interaction among factors.
Line 351: In social ecological approaches, time is an oft neglected dimension. this is especially true for the rapid rise in obesity and all of the factors that are interacting to increase obesity. A short note here and/or in conclusions might give some direction to future efforts.
Conclusions: some author suggestions regarding systems analysis and tools that can be used for future work would be great.
Author Response
Comment 1: This an exemplar of a systematic review; well organized and clear.
Response 1: Many thanks for your positive comments about our manuscript.
Comment 2: Line 42: clarify the ecological approach used with perhaps a reference to Bronfenbrenner and include some indication of the complexity of interaction among factors.
Response 2: As suggested, we have clarified (additional text underlined) that: “The ways in which non-parental childcare might affect obesity are poorly understood, may operate at a number levels within the socio-ecological model, and may interact in complex ways”. We have referenced a simple primer on the application of socio-ecological models to health research.
Comment 3: Line 351: In social ecological approaches, time is an oft neglected dimension. this is especially true for the rapid rise in obesity and all of the factors that are interacting to increase obesity. A short note here and/or in conclusions might give some direction to future efforts.
Response 3: As suggested, in the conclusion, we have added that: “Further work is required to clearly conceptualise proposed pathways linking childcare with diet and activity behaviours, and to determine whether, what aspects of, and how much exposure to childcare might discourage physical activity, and promote sedentary behaviour, less healthful diet and sleep patterns.”
Comment 4: Conclusions: some author suggestions regarding systems analysis and tools that can be used for future work would be great.
Response 4: As requested, in the conclusion we have added the sentence: “This may require wider thinking about the whole system of non-parental childcare and use of systems thinking – increasingly recognised as valuable to public health.”
Reviewer 2 Report
This manuscript is very extensive treatise about relationship between early childhood and factors which can influence occurrence obesity. Abstract has classics structure.
In the part of introduction we can learn about relationship to non-parental childcare. But I think that is missing an information about education structure, because each maternal school or "institutional child group" have some rules. It means that they have set requirements for daily meals, schedule activities during the day etc. For example in Czech republic are set Framework educational programs which clearly describe schedule activities (including activities physical). Also for schools is set which should be composition of daily meal. I think that this is important for understanding context of manuscript.
Methods has classic structure. Figure 1 is unnecessary.
In results:
The table 2 is very extensive (15 pages). I think that it can be very unclear for readers. It can be useful consider if it is necessary table or if it can be shorter or more clearly.
The table 3 is very extensive. It can be useful consider if it is necessary table or if it can be shorter or more clearly.
The manuscript is interesting and I think that it can be bring a lot of stimuli for practice. But it is very extensive and I think that his sense is lost mainly by stretching the text. It can be useful try to shorter text and highlight the main ideas.
Author Response
Comment 1: In the part of introduction we can learn about relationship to non-parental childcare. But I think that is missing an information about education structure, because each maternal school or "institutional child group" have some rules. It means that they have set requirements for daily meals, schedule activities during the day etc. For example, in Czech Republic are set Framework educational programs which clearly describe schedule activities (including activities physical). Also for schools is set which should be composition of daily meal. I think that this is important for understanding context of manuscript.
Response 1: As suggested, we have added that: “This is reflected in guidance and policies in some countries on physical activity and food in some childcare settings.” (lines 44-46) and provided example references.
Comment 2: Figure 1 is unnecessary.
Response 2: Figure 1 describes the eligibility criteria for the study and we believe it adds value to the manuscript. Reporting eligibility criteria is a core requirement of the PRISMA recommendations – the reporting guidance we used when developing our protocol and manuscript. It may be very difficult for readers to fully understand what our systematic review was without including the eligibility criteria in the figure. Whilst this information could be converted from a figure to text, the reviewer notes that the manuscript is already lengthy. Our preference is, therefore, to leave Figure 1 as is. We would happily convert it to text if the editor prefers.
Comment 3: The table 2 is very extensive (15 pages). I think that it can be very unclear for readers. It can be useful consider if it is necessary table or if it can be shorter or more clearly.
Response 3: Whilst we only found 13 studies meeting the eligibility criteria, many studies reported on numerous different measures of exposure coupled with numerous different measures of outcome – giving a total of 90 different associations. This explains the extensive nature of Table 2. Providing results of individual studies is another core requirement of the PRISMA recommendations. As such, we believe that Table 2 must be retained. If the editor prefers, this table could be moved to supplemental material.
Comment 4: The table 3 is very extensive. It can be useful consider if it is necessary table or if it can be shorter or more clearly.
Response 4: Table 3 summarises the results of applying the risk of bias tool selected to the included studies. It is less than 1.5 pages long. Again, providing this information is a requirement of the PRISMA recommendations and not something that we feel can be removed. However, if the editor prefers this table could be moved to supplemental material.
Comment 5: The manuscript is interesting and I think that it can be bring a lot of stimuli for practice. But it is very extensive and I think that his sense is lost mainly by stretching the text. It can be useful try to shorter text and highlight the main ideas.
Response 5: We have carefully reviewed the entire manuscript for length and reduced the main text length from 5389 words to 4866 words (excluding tables and figures) – a reduction of nearly 10%.
Round 2
Reviewer 2 Report
Authors of this manuscript done very good work. They adequate corrected it. I still think that manuscript is very long - mainly tables. I think that it can be confusing for reader, maybe can reduce interest of manuscript. It depend on editor decision if it is possible to publish in this version.
Author Response
We thank the reviewer for their comments.
In line with our previous response to the issue of length of the manuscript - this is due to the long tables (mainly Table 2 presenting included studies' results) which is a normal feature of systematic reviews and required by the PRISMA statement. If the Editor considers that moving Table 2 to a Supplement would improve the readibility of the manuscript, we would be happy to do this; however, our preference would be to have this table in the main body of text.